# Multifaceted Functions of TWSG1: From Embryogenesis to Cancer Development

**DOI:** 10.3390/ijms232112755

**Published:** 2022-10-22

**Authors:** Eri Suzuki, Tomohiko Fukuda

**Affiliations:** Department of Obstetrics and Gynecology, The University of Tokyo Hospital, Hongo 7-3-1, Bunkyo-ku, Tokyo 113-8655, Japan

**Keywords:** TWSG1, BMP, Chordin, dorsoventral axis formation, agonist, antagonist, follicular growth, cancer

## Abstract

Bone morphogenetic proteins (BMPs) play an important role in development. Twisted gastrulation BMP signaling modulator 1 (TWSG1) was initially identified as a regulator of the dorsoventral axis formation in Drosophila. The mechanism of BMP signaling modulation by TWSG1 is complex. TWSG1 inhibits BMP signaling by binding to BMP ligands including BMP4, whereas it enhances signaling by interacting with Chordin, a BMP antagonist. Therefore, TWSG1 can act as both a BMP agonist and antagonist. TWSG1 has various functions ranging from embryogenesis to cancer progression. *TWSG1* knockout mice showed neural, craniofacial, and mammary defects. TWSG1 also regulated erythropoiesis and thymocyte development. Furthermore, the relationship between TWSG1 and cancer has been elucidated. Allelic loss of *TWSG1* was detected in colorectal cancer. TWSG1 expression was upregulated in papillary thyroid carcinoma and glioblastoma but downregulated in gastric and endometrial cancers. TWSG1 suppressed BMP7-enhanced sphere formation and migration in endometrial cancer cells, indicating its tumor-suppressive role. Further studies are required to clarify the TWSG1 function and its association with BMP signaling in cancer development. Finally, TWSG1 is abundantly expressed in human and mouse ovaries and sustains follicular growth in rodent ovaries. Thus, TWSG1 has various functions ranging from fertility to cancer. Therefore, TWSG1 signaling modulation may be beneficial in treating specific diseases such as cancer.

## 1. Introduction

Bone morphogenetic proteins (BMPs) belong to the transforming growth factor β (TGFβ) superfamily. BMPs are secreted extracellular matrix-associated proteins that perform crucial roles in the morphogenesis of several organs, including the musculoskeletal, cardiovascular, and neural systems [1]. In addition, the association between BMP and cancer development has been elucidated in a variety of cancers including colorectal and ovarian cancer [1].

The twisted gastrulation BMP signaling modulator 1 (TWSG1) gene encodes a secreted protein, which modulates BMP signaling. TWSG1 is a secretory BMP-binding glycoprotein that regulates the activity of BMP ligands in the extracellular matrix and is associated with a variety of diseases [2]. TWSG1 is a homolog of the twisted gastrulation (TSG) gene, initially identified as a regulator of rigid dorsoventral axis formation in Drosophila [3]. Mice, Zebrafish, and Xenopus also have TSG homologs [4,5]. During dorsoventral gradient formation, Chordin is the primary BMP regulator in many organisms including Xenopus and Zebrafish [6,7]. Chordin antagonizes BMP signaling by binding to BMP ligands and preventing them from interacting with their cell surface receptors, and TSG is known to strengthen this inhibitory complex [8]. Chordin contains four cysteine-rich domains of about 70 amino acids each, mediating the direct binding of Chordin to BMP. The binding of BMP to Chordin prevents the binding of BMP to its cognate receptor, leading to the blockade of BMP signaling [9]. This regulation is modulated by the secreted glycoprotein TSG, which forms a ternary complex with Chordin and BMP. The cleavage sites of Chordin are located at the conserved aspartic acid residues just downstream of CR1 and CR3, cysteine-rich domains [10]. Besides the stabilization of the ternary complex [8], the cleavage of Chordin is also triggered by TSG, followed by Xolloid metalloprotease recruitment [10]. Once cleaved by Xolloid-dependent proteolysis, the individual CR1 and CR3 domains bind to BMP with 10-fold lower affinity than full-length Chordin, leading to BMP signaling activation. This mechanism is important in the regulation of Spemann organizer activity, the determinant of dorsal axis formation [10].

TSG homologs share 72–98% homology in amino acid sequences and retain 24 cysteine residues in Drosophila [5]. TSG expression is limited to early embryos in Drosophila, whereas its expression can be observed throughout embryonic development in mice and humans [3]. The correct specification of dorsal midline cells by TSG requires decapentaplegic (DPP), the Drosophila ortholog of bone morphogenetic proteins 2 and 4 (BMP2 and BMP4). DPP/BMP signaling is highly regulated via extracellular antagonists; Noggin and Chordin in vertebrates, short gastrulation, or Sog in Drosophila. Together with other members of the TGFβ superfamily of secreted proteins, BMPs control many aspects of development and homeostasis in a number of organisms including gastrulation and neural induction in vertebrates [11]. According to the Human Protein Atlas, TWSG1 is mainly produced from fibroblasts and is present throughout the body at both the mRNA and protein levels [12], suggesting an essential function of TWSG1 in the human body. TWSG1 modulates BMP signaling after cellular secretion. However, the mechanism of TWSG1-mediated BMP signaling modulation is complex since TWSG1 is able to function as both an agonist and antagonist of BMP signaling (Figure 1). Most BMP antagonists work independently by directly binding to their selective BMP ligands and as a result, this precludes their interaction with BMP receptors. Otherwise, it has been reported that the TSG protein in Drosophila and TSG homolog in Xenopus can go further to interact with Chordin, another BMP antagonist, by acting as a cofactor [13]. Similar to other BMP antagonists, TWSG1 inhibits BMP signaling by directly binding to BMP ligands including BMP4 [4,14,15] (Figure 1A). In addition, TWSG1 and the BMP endothelial cell precursor-derived regulator (BMPER), another BMP signaling modulator, synergistically suppress BMP4 signaling through a direct interaction [16]. TWSG1 also inhibits BMP7 signaling [8] but has a negligible effect on BMP2 signaling [17,18]. It represses BMP2/BMPR1A binding but not the BMP2/ACVR2B interaction [19]. Generally, both TSG homologs and full-length Chordin can synergistically bind BMPs to form a ternary complex; this stabilizes the interaction between Chordin and BMPs and thus greatly enhances the inhibitory ability of Chordin [13] (Figure 1A). These findings indicate that TWSG1 functions in a BMP ligand-dependent manner. In contrast, TWSG1 enhances BMP4 signaling through the formation of the BMP4–Chordin–TWSG1 ternary complex [20]. As previously described in TSG1, TWSG1 similarly triggers the dissociation of BMP4 and Chordin through the tolloid-dependent cleavage of Chordin, resulting in BMP-signaling activation [20] (Figure 1B). Therefore, TWSG1 can act as both a BMP agonist and antagonist.

BMP signaling is tightly regulated at multiple stages [1]. BMP ligands are synthesized and secreted as larger immature proproteins that are then cleaved by extracellular proprotein convertases such as Furin. Subsequently, mature BMPs are captured by several BMP antagonists including Noggin, Gremlin, and TWSG1. BMPs have many more antagonists compared to other TGFβ family ligands [1]. This antagonism has specificity. The distribution of BMP ligands and antagonists defines their signaling activity. Free BMPs form dimers that interact with BMP type I and type II serine/threonine kinase receptors forming a hexametric complex, followed by type I receptor activation [1,21] (Figure 1C). ACVRL1 (ALK1), ACVR1 (ALK2), BMPR1A (ALK3), and BMPR1B (ALK6) are classified as type I receptors and ACVR2A (ActRII), ACVR2B (ActRIIB), and BMPR2 (BMPRII) as type II receptors [21]. After type I receptor activation, SMAD1/5/8 are phosphorylated and form complexes with SMAD4, which are translocated to the nucleus, where they activate the transcription of target genes [21] (Figure 1C). ID family genes are well-known downstream targets. The expression of SNAIL and SLUG, EMT transcriptional factors, are also regulated by BMP signaling [18]. Although the molecular basis of the canonical SMAD pathway and its role in gene transcription is well explored, the molecular activation mechanism and the cellular functions of non-SMAD pathways, which rather act directly and independently of gene transcription, are poorly investigated. In particular, the molecular mechanism of BMP-induced phosphatidylinositol 3-kinase (PI3K) activation, its signaling route, and cellular function are poorly characterized (Figure 1C). In recent years, several studies have shown the requirement of PI3K for the BMP2-induced migration of various cell types of mesenchymal origin by as yet unknown mechanisms. Cell-surface heparan sulfates and extracellular biglycan are indispensable for this interaction [22,23]. BMPs can also trigger rapid cellular responses by activating several mitogen-activated protein kinases (MAPKs) such as extracellular signal-regulated protein kinase (ERK)1/2, c-Jun N-terminal kinase (JNK), and p38 MAPK [21] (Figure 1C). Although the downstream targets of the non-SMAD pathways are not clearly understood, BMP signaling has a variety of functions in development. Furthermore, the tumor-modulating effects of BMP signaling have also been elucidated in rodents [21].

Considering that BMP ligands and antagonists are distributed throughout the body, TWSG1 could have a wide range of functions. In this review, we focus on TWSG1 biology ranging from embryogenesis to cancer development (Figure 2).

## 2. TWSG1 and Embryogenesis

TSG, a TWSG1 homolog, regulates dorsoventral axis formation in Drosophila, fly, frog, and fish embryos [4]. The TSG gene is one of seven known zygotic patterning genes that specify the fate of dorsal cells in Drosophila embryos. Mutations at the twisted gastrulation (TSG) loci interfere with early morphogenetic movements in Drosophila. Because of the mutation, embryos can fully extend their germbands but they do not extend dorsally. As a result, when normal embryos have fully extended germbands, the germbands in mutant embryos are folded into the interior on the ventral side of the embryo. TSG-mutated embryos have abnormal deep dorsal folds during early gastrulation, resulting in the failure of dorsal cells to slip laterally to make way for the expanding germband. They could develop but end up forming disorganized first-instar larvae [3]. The combination of TSG and decapentaplegic (DPP), a BMP2/4 homolog, determines the dorsal midline cell fate in Drosophila [24,25]. Drosophila Sog, a Chordin homolog, is cleaved by the metalloprotease tolloid in the presence of TSG, leading to DPP signaling activation [26]. Xenopus TSG also disrupts the interaction between BMP4 and Chordin [22]. TSG also enhances BMP signaling in the absence of Chordin in Zebrafish embryos [27], suggesting an interaction between TSG and an unknown factor. Thus, TSG functions as a BMP agonist. In parallel, TSG acts as a BMP antagonist. Several groups have reported that TSG forms a ternary complex with BMP and Chordin, resulting in stabilizing the inhibitory complex [4,14,15]. The antagonistic activity of TSG toward BMP signaling is conserved between Drosophila and vertebrates [4,14]. However, the dual functions of TSG homologs have not yet been fully elucidated. Loss-of-function studies have revealed a variety of phenotypes. In Xenopus, TSG knockdown results in moderate head and forebrain defects, which can be rescued by Chordin overexpression and BMP depletion [28,29]. It also causes moderately strong dorsalization of the embryonic axis in Zebrafish [30]. In mice, TSG homozygous mutants display mild vertebral abnormalities and osteoporosis [31]. These mice also display widely variable craniofacial phenotypes that correlate with embryonic salivary gland dysmorphogenesis [32]. TSG null: BMP4 homozygous mutant mice are embryonically lethal and display holoprosencephaly (HPE), first branchial arch, and eye defects [31]. Furthermore, the simultaneous loss of TSG and BMP7 causes sirenomelia in mice [33]. However, TSG-null mice show trabecular bone defects at 4 weeks old, but normal phenotypes at 7 weeks old, suggesting a stage-specific function of TSG in embryogenesis [34].

## 3. TWSG1 and Neurogenesis

TWSG1 has previously been reported to be expressed in multiple tissues during the gastrulation, neurulation, and organogenesis stages such as the neural plate, neural tube, somite, branchial arches, surface ectoderm, choroid plexus, hippocampus, and endoderm. TWSG1-mutant mice exhibit neural arch deletions in the cervical vertebrae and show a high mortality rate of up to about 50% [35]. As previously described, TWSG1-null BMP4 homozygous mutant mice showed HPE, characterized by the failure of the prosencephalon to undergo median cleavage into bilateral cerebral hemispheres [31]. TWSG1 has been said to enhance BMP signaling by promoting the degeneration of Chordin. Although 7% of wild-type mice exposed to retinoic acid developed HPE, all TWSG1-deficient mice exposed to retinoic acid had severe HPE [36]. HPE in humans has been associated with chromosomal deletions of 18p11.3, containing the *TWSG1* gene [14]. However, *TWSG1* mutations have rarely been detected in human HPE [37]. Among all other regions of the brain such as the hippocampus, the choroid plexus expresses the highest TWSG1 level in both mice and humans. It is believed that TWSG1 is the main regulator of BMP present in the choroid plexus, which plays an important role in neural progenitor differentiation and neuronal repair [38]. In addition, TWSG1-null mice have shown a higher frequency of hydrocephalus than wild-type mice [38], suggesting a neuroprotective function of TWSG1.

## 4. TWSG1 and Osteogenesis

Because BMP signaling is indispensable for osteogenesis [39], TWSG1 may play an essential role in this process. In fact, TWSG1-null mice exhibit mild vertebral abnormalities and osteoporosis [31]. TWSG1 knockout mice had a significant delay in the ossification of cervical and coccygeal vertebrae in newborns. As result, they were smaller in size than the wild-type mice [35]. Interestingly, craniofacial defects in TWSG1-null mice were significantly rescued by p53 deletion [40]. The reduction in the absolute number of osteoblasts explains the decreased serum levels of osteocalcin observed in TSG-null mice. TWSG1-deficient mice also display dwarfism with delayed endochondral ossification [41]. Moreover, osteoblasts isolated from TWSG1-null mice show impaired BMP signaling [34]. Therefore, TWSG1 acts as a BMP agonist during skeletogenesis. Enhanced osteoclastogenesis was also observed in TWSG1-null mice. Whether TSG favors or opposes BMP activity in osteoblasts is related to its levels of expression, as has been reported for non-osteoblastic cells [34]. These results indicate that the TSG gene inactivation causes a transient decrease in bone volume, associated with a decrease in bone mineral content and serum osteocalcin levels, without changes in the osteoblast or osteoclast number per surface or activity [34]. In contrast to osteoblasts, osteoclasts isolated from TWSG1-null mice displayed BMP signaling activation [34], implying the BMP antagonistic activity of TWSG1 in osteoclastogenesis. Consistently, TWSG1 overexpression has resulted in the inhibition of osteoclastogenesis in primary osteoclasts [42], and this inhibition is dependent on BMP binding [43]. TWSG1 can also be found in cartilage, joints, and tendons [44]. TWSG1 inhibits BMP2-stimulated SMAD1 phosphorylation in chondrocytes, and TWSG1 overexpression suppresses cartilage development in vivo [45].

## 5. TWSG1 and Hematopoiesis

TWSG1 plays an important role in T-cell differentiation and erythropoiesis. Both T and B cells dynamically express TWSG1 [46]. TWSG1-deficient mice display an atrophic thymus [41], which cooperates with Chordin to inhibit the BMP4-mediated suppression of thymocyte growth and differentiation [47]. TWSG1 suppressed hepcidin indirectly by inhibiting the signaling effects and associated hepcidin upregulation by BMP2 and BMP4 [48]. TWSG1 mRNA expression is enhanced after T-cell activation, leading to strong suppression of proliferation and cytokine production in primed alloreactive CD4+ T cells [49]. Surprisingly, this suppression depends on TGFβ signaling but not BMP signaling [49]. TWSG1 also regulates B-cell function. B cells express the BMP modifier TWSG1 in an activation-dependent manner. B-cell-specific TWSG1 deficiency does not affect B-cell development but it does alter B-cell responses. TWSG1 is a modulator of BMP activity, and no function for TWSG1 without BMP interaction has been reported. In the search for a BMP signaling network that could be regulated by TWSG1 in situ, BMP2 was identified in the vicinity of the splenic marginal zone ring [50]. In thalassemic mice with ineffective erythropoiesis, increased TWSG1 expression was observed in the spleen, bone marrow, and liver compared to that in healthy mice [48]. TWSG1 repressed BMP-inducible hepcidin, which positively regulated erythropoiesis [48]. Therefore, TWSG1 inhibits erythropoiesis.

## 6. TWSG1 and Angiogenesis

The previous report showed that the BMP endothelial cell precursor-derived regulator (BMPER), one of the extracellular BMP modulators, acts proangiogenically on endothelial cells in a concentration-dependent manner. In addition, TWSG1 enhances endothelial cell ingrowth, migration, and sprouting, as confirmed by a mouse Matrigel plug and human umbilical vein endothelial cells (HUVEC) sprouting assay [51]. Accordingly, TWSG1 secreted from papillary thyroid cells induces HUVEC sprouting [52]. The expression of a panel of angiogenic factors in the tumor cell culture supernatant in the absence of TWSG1 showed that vascular endothelial growth factor (VEGF) was significantly decreased, whereas endostatin, PAI-1, and thrombospondin-1 were increased in the absence of TWSG1 compared to the control [52].

TWSG1 induces the phosphorylation of SMAD1/5/8, AKT, and extracellular signal-regulated kinase (ERK) [52], indicating that TWSG1 is a BMP agonist in endothelial cells. Interestingly, TWSG1 silencing also augments HUVEC sprouting [51], suggesting a function other than that of a secreted protein. In addition, TWSG1 and BMPER interfere with each other to function as pro-angiogenic factors [51]. This interaction is also important for arterial-venous specification, caudal vein plexus (CVP) formation, and correct development in Zebrafish [53].

## 7. TWSG1 and Cancer Development

According to the Human Protein Atlas (https://www.proteinatlas.org/ accessed on 11 October 2022), TWSG mRNA levels are different in various cancers (Figure 3). The relationship between TWSG1 expression and cancer development has also been elucidated (Figure 4). A loss of heterozygosity and truncating variants of *TWSG1* have been observed in familial colorectal cancer [54]. TWSG1 expression is upregulated in cholangiocellular carcinoma, hepatocellular carcinoma, papillary thyroid cancer (PTC), and glioblastoma but downregulated in gastric and endometrial cancers [2,18,52,55,56]. In gastric cancer, TWSG1 expression was low, and the lower expression level showed a correlation with higher pathological grading or the clinical stage of patients [2]. Thus, it is possible that TWSG1 acts as a tumor suppressor gene in gastric cancer. In PTC, lymph node metastases are correlated with high *TWSG1* mRNA expression [52]. Moreover, TWSG1 knockdown inhibits the proliferation, migration, and invasion of PTC cells. TWSG1 may promote the expression of matrix metalloproteinases (MMPs) and enhance the activity of the BMP pathway. Thus, TWSG1 could be a new diagnostic target for PTC [52]. TWSG1 also positively modulates glioblastoma cell growth in vitro and in vivo. The proliferation of glioma cells is somehow regulated by the PI3K/AKT pathway mentioned in Figure 1C. Patients with high expression levels of TWSG1 had a shorter time until recurrence than patients with low expressions of TWSG1. However, there was no correlation between TWSG1 expression and overall survival [56]. Furthermore, TWSG1 is highly expressed in cholangiocellular and hepatocellular carcinoma, and interestingly, immunohistochemistry and immunoblotting showed a stronger TWSG1 expression in cholangiocellular carcinoma than in hepatocellular carcinoma. The difference in expression was significant. In addition, a striking correlation between TWSG1 and BMP4 expression was found in cholangiocellular carcinoma, which raises questions about the possible role of TWSG1 in modulating invasive behavior or initiating the process of lymphovascular invasion. In contrast, TWSG1 suppresses BMP7-enhanced sphere formation and migration in endometrial cancer cells [10], indicating the tumor-suppressive role of TWSG1 in endometrial cancer. Furthermore, TWSG1 expression is downregulated by promoter methylation in human malignant mesothelioma side-population (SP) cells compared to non-SP cells [57]. Therefore, TWSG1 may influence cancer stemness in malignant mesothelioma. Further studies are needed to clarify the function of TWSG1 and its association with BMP signaling in cancer development.

## 8. TWSG1 and Folliculogenesis

TWSG1 expression was initially detected in adult sheep ovaries [58]. Juvenile mouse ovaries also exhibited consistently high TWSG1 expression, which was not affected by FSH exposure in the granulosa cells of developing follicles [59]. Wang et al. found that TWSG1 and Nbl1 were the most abundant BMP antagonists expressed in rodent and human ovaries [60]. TWSG1 has been reported to have synergistic action with the Chordin subfamily, including CHRD and CHRDL1, the genes of which also showed moderate expression in the mammalian ovary. Bioactivity tests indicated that TWSG1 alone can directly inhibit the signaling of BMP6 or BMP7 by masking the receptor binding sites of BMPs. BMP2, BMP4, BMP7, GDF5, and activin A have been said to have an antagonizing ability towards CHRD, which has synergic effects with TWSG1 [59]. TWSG1 is the most abundant BMP antagonist in mouse ovaries [60]. TWSG1 colocalizes with Chordin within the theca cells, whereas it colocalizes with Chordin-like 1 in the developing granulosa cells of mouse ovaries [60], suggesting the differential roles of TWSG1 in each ovarian compartment. Some data suggest that TWSG1 may work coordinately with CHRD within theca/interstitial shells and also with CHRDL1 in developing granulosa cells. These interactions would modulate the intraovarian functions of the TGFβ superfamily members such as the control of progesterone production [59].

## 9. Other Functions of TWSG1

TWSG1 is also indispensable for mammary gland development in mice. Regulation of BMP signaling by TWSG1 is required for normal ductal elongation, branching of the ductal tree, lumen formation, and myoepithelial compartmentalization in the postnatal mammary gland [61]. Understanding how TWSG1 regulates normal ductal maturation could provide insights into the role it may be playing during carcinogenesis. Moreover, TWSG1 modulates BMP signaling following kidney injury. BMP7 plays an important role in establishing embryonic nephrons, protecting them from acute renal injuries and preventing and reversing chronic kidney injuries. For example, BMP7 counteracts and reverses epithelial–mesenchymal transition in renal epithelia by regulating E-cadherin. BMP7 also reduces the proximal tubule expression of cytokines such as IL-6 and macrophage chemoattractant protein, modifying inflammatory infiltration. Healthy kidneys express BMP7-antagonistic CHRDL1 in the straight segment of the proximal tubule and demonstrate little active BMP signaling in this region [62]. When a renal injury occurs, the epithelium of the straight segment is sloughed off and CHRDL1 activity is lost, correlating with increased BMP’s recovering signal response [62]. Finally, CHRDL1 expression is regained upon differentiation of regenerated epithelia, and BMP signaling is again reduced to basal levels [62]. Similarly, kidney injury triggers TWSG1 downregulation, leading to BMP signaling activation and renal protection [63].

## 10. TWSG1 and Future Perspectives

Although TWSG1 plays important roles in many aspects, the mechanism of how TWSG1 is regulated remains to be elucidated. Once the regulation mechanism is understood, we could suppress the progression of cancers and specific diseases by modulating its expression. Considering its abundant expression in ovaries, TWSG1 could be a new marker of fertility. In addition, recombinant TWSG1 has an advantage over conventional BMP receptor kinase inhibitors in that it has no effect on other receptors. Therefore, recombinant TWSG1 could be used as a novel BMP inhibitor.

## 11. Conclusions

In conclusion, TWSG1 has a variety of functions ranging from fertility to cancer in multiple organs. However, the detailed mechanisms underlying TWSG1 functions have not yet been fully elucidated. Although further studies are needed to clarify these mechanisms, the modulation of TWSG1 signaling could be promising for treating specific diseases and cancers.

## Figures and Tables

**Figure 1 ijms-23-12755-f001:**
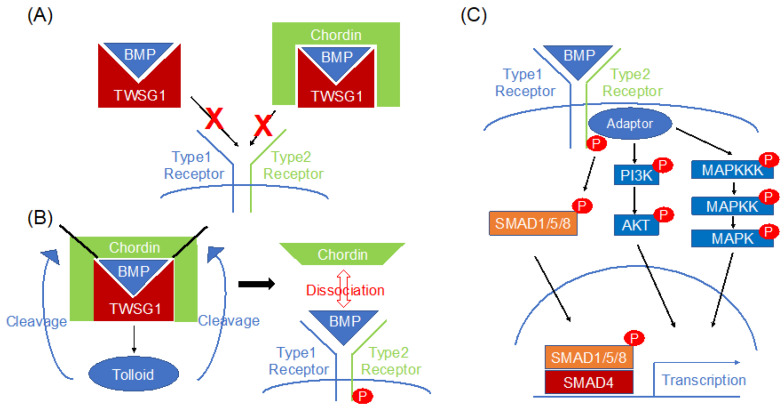
TWSG1 can function as both a BMP agonist and an antagonist. (**A**) TWSG1 functions as a BMP antagonist by inhibiting BMP signaling through BMP ligand binding. TWSG1 also represses BMP signaling through the formation of the BMP–TWSG1–Chordin ternary complex. (**B**) TWSG1 also functions as a BMP agonist. TWSG1 triggers BMP/Chordin dissociation via the tolloid-dependent cleavage of Chordin. (**C**) Overview of BMP signaling. After activating the receptors, phosphorylated SMAD1/5/8 translocate to the nucleus with SMAD4, leading to transcriptional activation. BMPs can also activate non-SMAD pathways, including PI3K/AKT and MAPK pathways.

**Figure 2 ijms-23-12755-f002:**
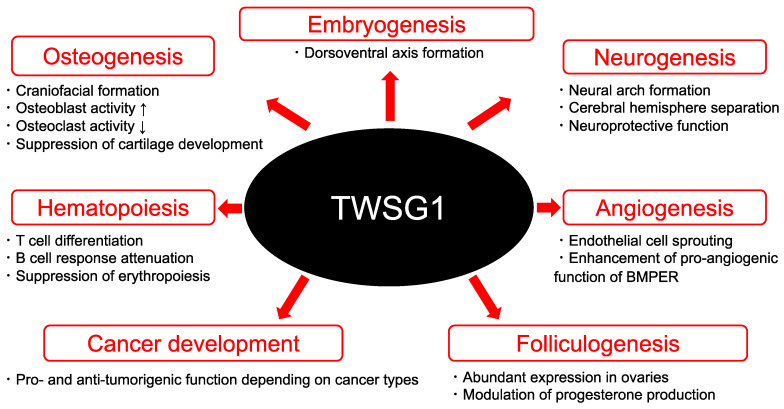
The multifaceted functions of TWSG1. TWSG1 has a wide range of functions from embryogenesis to cancer development.

**Figure 3 ijms-23-12755-f003:**
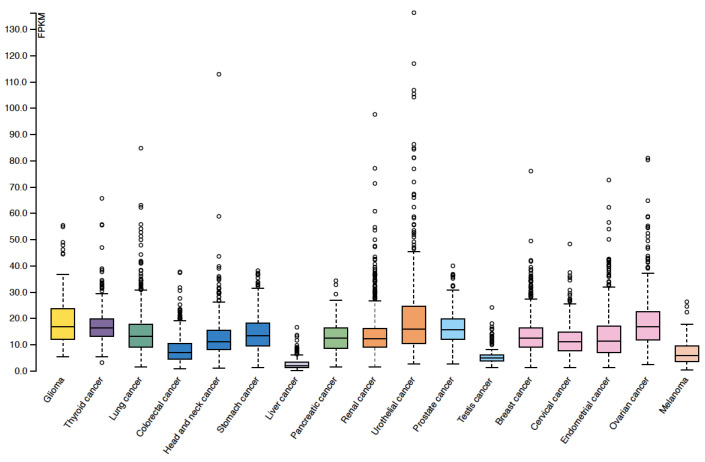
TWSG1 mRNA expression in various types of cancers. The mRNA expression profiles of TWSG1 in various types of cancers were acquired from the Human Protein Atlas (https://www.proteinatlas.org/ accessed on 11 October 2022).

**Figure 4 ijms-23-12755-f004:**
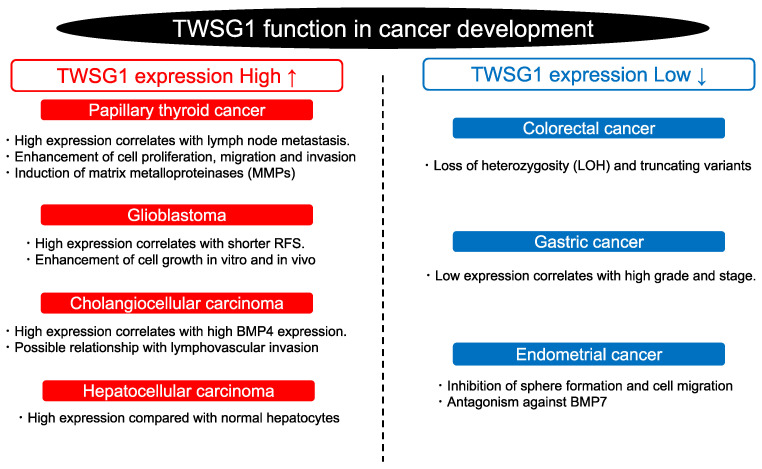
TWSG1 function in cancer development. TWSG1 has a variety of functions depending on the type of cancer. RFS (relapse-free survival).

## Data Availability

The expression profiles of TWSG1 were acquired from the Human Protein Atlas (https://www.proteinatlas.org/ accessed on 11 October 2022).

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
