# Peer review of "Multifaceted Functions of TWSG1: From Embryogenesis to Cancer Development"

_ijms, 2022, doi:10.3390/ijms232112755_

Round 1

Reviewer 1 Report

In this interesting review the authors discuss the various roles of Twisted 8 gastrulation BMP signaling modulator 1 (TWSG1),  a modulator of Bone morphogenetic proteins (BMP) signalling that can be both agonist and antagonist the authors discuss the roles of TWSG1 on various physiological pathways, but also the relationship between TWSG1 and cancer.

Mjor concern : this review should ahve at least two more recapitulative figures, one summarizing the roles of TWSG1 on various physiological pathways, the other on the relationship between TWSG1 and cancer

Minor :

holoprosencephaly is cited for the first time line , yet abbrevaition is in line 171.

line258, the sentence "However, it had no correlation between over 258 survival " has no clear meaning.

Author Response

Details of response to reviewer1:

 We highly appreciate the editor and the reviewers for giving us the opportunity to revise our manuscript. We have repeated the comments of the reviewers and have responded to the comments.

Reviewer 1

Evaluation

1) In this interesting review the authors discuss the various roles of Twisted 8 gastrulation BMP signaling modulator 1 (TWSG1), a modulator of Bone morphogenetic proteins (BMP) signalling that can be both agonist and antagonist the authors discuss the roles of TWSG1 on various physiological pathways, but also the relationship between TWSG1 and cancer.

Response

Thank you for your favorable comments.

2) Mjor concern : this review should ahve at least two more recapitulative figures, one summarizing the roles of TWSG1 on various physiological pathways, the other on the relationship between TWSG1 and cancer

Response

As suggested by the reviewer, we have modified figure 2 and table 1 to improve readers’ understanding.

Please see figure 2 and figure 3 of the revised manuscript.

3) Minor :

holoprosencephaly is cited for the first time line , yet abbrevaition is in line 171.

line258, the sentence "However, it had no correlation between over 258 survival " has no clear meaning.

Response

As suggested by the reviewer, we have modified the description.

Please see page 4, line 161 and page 6, lines 258-259 of the revised manuscript.

Reviewer 2 Report

In this review Suzuki et al. have tried to comprehensively describe the multifaceted role of TWSG1 in different developmental processes and cancer. Although this work might be interesting to a specific group of researchers but the review needs to be improved in multi aspects.

1. Throughout the review the role of TWSG1 has been mentioned in various aspects. However the mechanism by which it acts in different processes has not been mentioned.

2. Instead of mentioning repeatedly about the expression of TWSG1 in various cancer authors should provide a bar graph or box plot to show the expression of TWSG1 in various cancers.

3. It might be interesting to have a table showing the brief function of TWSG1 in various cancer/developmental process.

4. Authors should include a section which emphasizes on the importance/biological significance of TWSG1 in clinic and how can it have a great impact in this field in the near future. 

Author Response

Details of response to reviewer2:

 We highly appreciate the editor and the reviewers for giving us the opportunity to revise our manuscript. We have repeated the comments of the reviewers and have responded to the comments.

Reviewer 2

Evaluation

In this review Suzuki et al. have tried to comprehensively describe the multifaceted role of TWSG1 in different developmental processes and cancer. Although this work might be interesting to a specific group of researchers but the review needs to be improved in multi aspects.

Response

Thank you for your valuable comments.

1) Throughout the review the role of TWSG1 has been mentioned in various aspects. However the mechanism by which it acts in different processes has not been mentioned.

Response

Unfortunately, the detailed mechanism of TWSG1 function in different processes has not been fully elucidated since most previous reports focused on phenotypes of knockout organisms. However, we summarized BMP agonistic and antagonistic functions of TWSG1 in Figure 1. Considering the kinetics of expression, concentration gradients of TWSG1, BMPs and Chordin may define its function.

2) Instead of mentioning repeatedly about the expression of TWSG1 in various cancer authors should provide a bar graph or box plot to show the expression of TWSG1 in various cancers.

Response

To improve reader understanding, we have turned table 1 into figure 3.

It is interesting to compare TWSG1 expression between different cancers. However, it is impossible because each study assessed TWSG1 expression at different levels e.g. qPCR and immunohistochemistry.

Please see figure 3 of the revised manuscript.

3) It might be interesting to have a table showing the brief function of TWSG1 in various cancer/developmental process.

Response

As suggested by the reviewer, we have modified figure 2 and table 1.

Please see figure 2 and figure 3 of the revised manuscript.

4)  Authors should include a section which emphasizes on the importance/biological significance of TWSG1 in clinic and how can it have a great impact in this field in the near future. 

Response

As suggested by the reviewer, we have added the description.

Please see page 8, lines 312-319 of the revised manuscript.

Round 2

Reviewer 2 Report

Comment2: Authors can use RNA-seq data from Human protein atlas to compare the expression of TWSG1 in various cancer.

Authors have satisfactorily responded to all other comments.

Author Response

Details of response to reviewer2:

 We highly appreciate the editor and the reviewer 2 for giving us the opportunity to revise our manuscript. We have repeated the comment of the reviewer 2 and have responded to the comment.

Reviewer 2

Evaluation

Comment2: Authors can use RNA-seq data from Human protein atlas to compare the expression of TWSG1 in various cancer.

Response

Thank you for your valuable suggestion.

As suggested by the reviewer, we added RNA-seq data from the Human Protein Atlas as figure 3.

Please see figure 3 and page 6, lines 243-244 of the revised manuscript.
